# Electrospun Scaffolds Containing Silver-Doped Hydroxyapatite with Antimicrobial Properties for Applications in Orthopedic and Dental Bone Surgery

**DOI:** 10.3390/jfb11030058

**Published:** 2020-08-14

**Authors:** Thomas E. Paterson, Rui Shi, Jingjing Tian, Caroline J. Harrison, Mailys De Sousa Mendes, Paul V. Hatton, Zhou Li, Ilida Ortega

**Affiliations:** 1School of Clinical Dentistry, University of Sheffield, Shefield 0114, UK; t.paterson@sheffield.ac.uk (T.E.P.); c.j.harrison@sheffield.ac.uk (C.J.H.); i.ortega@sheffield.ac.uk (I.O.); 2Beijing Laboratory of Biomedical Materials, Institute of Traumatology and Orthopaedics, Beijing Jishuitan Hospital, Beijing 100083, China; shirui@jst-hosp.com.cn; 3Beijing Key Laboratory of Micro-nano Energy and Sensor, Beijing institute of Nanoenergy and Nanosystems, Chinese Academy of Sciences, Beijing 100083, China; tianjingjing@pumch.cn; 4Central Laboratory, Peking Union Medical College Hospital, Peking Union Medical College and Chinese Academy of Medical Sciences, Beijing 100730, China; 5Certara, Simcyp division, Sheffield 0114, UK; mailys.mendes@certara.com

**Keywords:** electrospinning, antimicrobial, nano-hydroxyapatite, silver, toxicity, bone regeneration

## Abstract

Preventing the development of osteomyelitis while enhancing bone regeneration is challenging, with relatively little progress to date in translating promising technologies to the clinic. Nanoscale hydroxyapatite (nHA) has been employed as a bone graft substitute, and recent work has shown that it may be modified with silver to introduce antimicrobial activity against known pathogens. The aim of this study was to incorporate silver-doped nHA into electrospun scaffolds for applications in bone repair. Silver-doped nHA was produced using a modified, rapid mixing, wet precipitation method at 2, 5, 10 mol.% silver. The silver-doped nHA was added at 20 wt.% to a polycaprolactone solution for electrospinning. Bacteria studies demonstrated reduced bacterial presence, with *Escherichia coli* and *Staphylococcus aureus* undetectable after 96 h of exposure. Mesenchymal stem cells (MSCs) were used to study both toxicity and osteogenicity of the scaffolds using PrestoBlue^®^ and alkaline phosphatase (ALP) assays. Innovative silver nHA scaffolds significantly reduced *E*. *coli* and *S. aureus* bacterial populations while maintaining cytocompatibility with mammalian cells and enhancing the differentiation of MSCs into osteoblasts. It was concluded that silver-doped nHA containing scaffolds have the potential to act as an antimicrobial device while supporting bone tissue healing for applications in orthopedic and dental bone surgery.

## 1. Introduction

Preventing deep bone infections after surgery and enhancing bone regeneration reduces the time required for patient recovery and the costs associated with long hospital stays. Deep bone infections are very challenging to treat once established due to the difficulty of achieving a suitable antibiotic concentration in the affected area and ensuring all bacteria have been removed. Bacteria such as *Staphylococcus aureus* (*S. aureus*) and *Escherichia coli* (*E. coli*) are responsible for causing bone infections post-surgery [1,2], and the application of excessive antibiotics can lead to antimicrobial resistance. Antibacterial poly(methyl methacrylate) beads have been used routinely for treating infections in a reactive manner with additional disadvantages of requiring surgical removal after the treatment ends [3]. Silver has often been used to successfully treat bacterial and fungal infections [4,5] and can be delivered in a localized area. Metallic implants coated with silver have been investigated to prevent infections [6]. However, silver by itself does not enhance wound healing and can have a toxic effect on mammalian cells at higher concentrations [7,8].

To facilitate bone healing, surgeons currently employ hydroxyapatite in granular or paste form to stimulate bone regrowth. Hydroxyapatite is the mineral component of bone and can be manufactured synthetically and has been demonstrated to aid in bone regeneration [9]. Nanoscale hydroxyapatite (nHA) is an increasingly popular biomaterial due to its similarity to the hydroxyapatite in human bone and tooth enamel. Our group recently produced a bioinspired nHA material through a rapid mix preparation, which is slightly calcium deficient to better mimic biological apatite [10]. Using nHA-based medical devices to increase the regeneration rate of the bone after surgery could greatly decrease patient healing times. Unfortunately, despite its success in aiding bone regeneration, hydroxyapatite has no innate antimicrobial properties to prevent infection. Moreover, the implantation of medical devices provides an increased risk of bacterial colonization and eventual biofilm formation if left untreated [11]. Once a biofilm has been established it is very difficult to treat using antibiotics and even surgical intervention involves a high risk of reoccurrence [12].

There is growing interest in using a combination of silver and hydroxyapatite to produce a material with both antimicrobial/antifungal [13,14] and bone-regenerating properties [15,16,17,18]. Jiang et al. [19] produced foamed polyurethane scaffolds containing up to 40 wt.% silver phosphate particles (Ag_3_PO_4_) and nHA. These mold-formed scaffolds were porous and eluted silver for up to 22 days without toxic effects on the human osteosarcoma cell line MG63. Saravanan et al. [20] produced scaffolds using the freeze-drying technique incorporating nano-hydroxyapatite with nanoparticles of silver. Wilcock et al. [21] developed a novel, rapid mixing, wet precipitation method of producing bioinspired, nanoscale hydroxyapatite crystals doped with silver for antimicrobial action. A degradable scaffold could be placed against existing implants or used to hold packed bone substitutes (e.g., bone chips) into a bone defect.

The biomaterials’ research community has been keenly interested in using electrospinning as a containment system for ceramic-derived materials including hydroxyapatite [22]. Electrospinning is a technique for producing fibrous scaffolds that roughly resemble the body’s extracellular matrix. These can be made porous to allow cell thoroughfare or nanoscale to act as a barrier to cells, fulfilling a wide variety of applications. Ciraldo et al. [23] investigated three different manufacturing approaches using mesoporous glasses containing silver. These were glass coatings prepared using electrospraying, sol-gel methods, and directly spinning with polycaprolactone (PCL). For electrospinning, the authors were limited to 5% mesoporous glasses in PCL but did not investigate antimicrobial action. However, a recent paper by the same group demonstrated that the mesoporous glasses loaded with silver were antimicrobial [24]. Anjaneyulu et al. [25] manufactured doped hydroxyapatite through the sol-gel method and electrospun with poly(vinyl alcohol). Schneider et al. [26] electrospun tricalcium phosphate nanoparticles into poly(lactic-co-glycolic acid) (PLGA) scaffolds, looking at a single concentration of silver, but did not investigate mammalian toxicity of the scaffold. Furtos et al. [27] electrospun a PCL scaffold to contain both nHA and amoxicillin, which they tested against four bacterial strains and found that increasing the nHA content reduced the loading efficiency of the antibiotic.

A variety of polymers and calcium compounds have been used for producing these antimicrobial scaffolds. Different manufacturing technologies have also been employed to produce these as scaffolds. A relatively small number of the above publications have reported testing their material with primary mesenchymal stem cells (MSCs), which are key to understanding bone-healing mechanisms. Few of the published reports have investigated microbiological testing beyond agar diffusion assays with no time course elution antimicrobial testing.

Here we present the fabrication of novel, dual-function scaffolds that combine silver to impart antimicrobial properties and hydroxyapatite to enhance bone regeneration. Our laboratory previously developed a silver-doped nano-hydroxyapatite material formed by the wet precipitation method [21]. Here we show the incorporation of this innovative material into an electrospun scaffold alongside a thorough investigation of metal ion release, antimicrobial activity against clinically relevant bacterial strains, and in vitro biocompatibility using cultured mammalian MSCs. This research has provided an important proof-of-concept study that has the potential to underpin the development of a new generation of dual-action medical devices to address the growing challenges of deep bone infection and antimicrobial resistance.

## 2. Materials and Methods

### 2.1. Silver Nano-Hydroxyapatite (Ca_10_(PO_4_)_6_(OH)_2_) Manufacture via Wet Precipitation

This was based on our previously published methods [10,21]. In brief, calcium hydroxide (Sigma Aldrich, Dorset, UK) (50 mmol) was suspended in 500 mL deionized water (dH_2_O) into which 0, 1, 2.5, or 5 mmol silver nitrate (Sigma Aldrich, UK) (corresponding to 0, 2, 5, or 10 mol%, respectively) was added and stirred at 400 rpm for 1 h on a hotplate (Fisher Scientific, Loughborough, UK) at 90 °C. Phosphoric acid (Sigma Aldrich, UK) (30 mmol) was dissolved in 250 mL dH_2_O, poured into the calcium and silver preparation, and stirred for a further hour. The suspension was left to settle overnight, after which the supernatant was poured off and the silver-doped nHA suspension was washed with dH_2_O (3 × 500 mL). The silver-doped nHA suspensions were dried at 60 °C and ground in an agate mortar and pestle.

### 2.2. Imaging the Silver Nano-Hydroxyapatite and Electrospun Fibers Using SEM

Dried nHA powder or a 5 × 5 mm section of electrospun scaffold was attached to a carbon tab and sputter gold coated and then imaged using a TESCAN VEGA3 SEM (Tescan, Brno, Czech Republic) with an accelerating voltage of 15 kV.

### 2.3. Imaging the Silver Nano-Hydroxyapatite Using Transmission Electron Microscopy (TEM) and Energy-Dispersive X-Ray Spectroscopy (EDX)

The nHA powder (0.1 g) was dispersed in ethanol in an ultrasonic bath (ThermoFisher, Shanghai, China) for 1 h and a drop was placed onto a copper mounting grid with carbon film and allowed to dry under ambient conditions. Samples were then imaged on a transmission electron microscope (Tecnai G2 F30, ThermoFisher, Shanghai, China) under standard imaging and EDX detecting elements Ca, P, Ag, and O. EDX settings used were 20 kV, takeoff 14.8, amp time 7 µs, and 126.5 eV resolution.

### 2.4. Analyzing Crystal Phases Present Using X-Ray Diffraction (XRD)

Phase analysis was performed on powdered samples using XRD (STOE IP, Darmstadt, Germany) with Cu Kα λ = 1.5418 Å radiation to determine the crystal structure of the materials. The diffractometer was operated at 40 kV and 35 mA, with a 2θ range of 20–50°. The following The International Centre for Diffraction Data Powder Diffraction (ICDD-PDF) cards were used for phase identification: 9-432 hydroxyapatite, 6-505 silver phosphate.

### 2.5. Electrospinning Polycaprolactone (PCL) and Incorporating the Ag–nHA

Polycaprolactone (average 80,000 Mn, Sigma, UK) was dissolved in a specific solvent and then propelled by an electric field onto a collector where polymer fibers formed as the solvent evaporated. Adding both the silver-doped and undoped nHA powder to the solution at 20 wt.% allowed it to be incorporated into the electrospun scaffold. PCL (1.5 g) was dissolved into a solvent mixture (90% Dichloromethane [DCM], 10% Dimethylformamide [DMF]) (Sigma Aldrich, UK) and stirred using a magnetic stirrer bar (5 mm bar, Sigma Aldrich, UK) for 2 h. The nHA powders were incorporated at 20 wt.% of the PCL component and added into the dissolved PCL solution. The polymer-ceramic-solvent mixtures were placed in an ultrasonic bath for 30 min, before stirring using a magnetic stirrer bar for 30 min to ensure even distribution of powder in the solution. A 1-mL Luer-Lok syringe (BD, EU) was loaded with 1 mL of the PCL/nHA solution and this was electrospun using a custom electrospinning rig described previously [28]. In total, 2 mL of solution at 1 mL/hr was used to fabricate each scaffold at 17 kV with a spinning gap between the needle and collector of 20 cm.

### 2.6. Accelerated Degradation Study of Electrospun Scaffolds and Release Profile

To investigate the rate at which nHA was released during scaffold degradation, an accelerated degradation study was performed over 30 days. Samples were weighed and added to a 5-mL solution of 0.1 M NaOH (Sigma Aldrich, UK) in dH_2_O and placed in an oven at 37 °C. At each time point (days 0, 1, 7, 14, 21, and 28) material was also weighed and 0.5 mL of solution was removed, which was replaced with fresh solution. The solutions obtained were diluted to 5 mL using dH_2_O.

### 2.7. Inductively Coupled Plasma (ICP) Measurement of Material and Degradation Product

To ascertain the presence and abundance of various elements within the sample, ICP was utilized. Solid samples (e.g., 0.25 g–0.5 g) were weighed into a glass tube and 9 mL of concentrated hydrochloric acid (Sigma Aldrich, UK) added and 3 mL of concentrated nitric acid (Sigma Aldrich, UK) added. The samples were then placed into the heating block and the temperature was gradually raised to 150 °C and maintained at this temperature for 30 min. After removal from the block and cooling, the samples were diluted to 50 mL with 1% nitric acid. Samples were diluted down to 1 in 100 for analysis. Samples were analyzed using an ICP-OES (Spectro-Ciros-Vision Optical Emission Spectrometer (Spectro, Kleve, Germany) using a procedure with a detection limit of at least 10 ppm.

### 2.8. Scaffold Preparation for Cell Culture

Disks of each electrospun scaffold were prepared and sterilized prior to culture with cells. Using a 13 mm cork borer, disks (13 mm) of each scaffold were punched out of the primary collector sheet. On the day of use, the culture scaffolds were disinfected with 70% ethanol (Sigma Aldrich, UK) in dH_2_O for 30 min and then washed three times in dH_2_O for 5 min.

### 2.9. Contact Bacterial Toxicity Test

Antimicrobial activity was investigated using a classical diffusion agar assay. Isolates of *E. coli* and *S. aureus* were purchased from the China Center of Industrial Culture Collection (CICC, Beijing, China): *E. coli*, CICC23657 and *S. aureus*, CICC10384. The bacterial suspensions at stationary phase were firstly obtained by culture in LB (Luria-Bertani) medium overnight at 37 °C in a 120-rpm shaker. Then 1 mL of the *E. coli* and *S. aureus* suspensions were pipetted from the overnight phase into another 100-mL flask containing 50 mL of fresh LB. The mixture was then cultured at 37 °C for another 6 h to obtain bacterial suspensions at exponential growth phase. After that, the bacterial suspension was centrifuged at 8000 rpm, washed, and resuspended in sterilized physiological saline to reach a concentration of 1 × 10^7^ colony forming units (CFU)/mL. Then, 0.1 mL of the bacteria suspension with 1 × 10^7^ CFU/mL was pipetted and plated on the agar plate uniformly. Three disks of the electrospun material (12 mm diameter) were placed on each Petri dish. These were incubated at 37 °C for 24 h. The zone of inhibition ring around the electrospun scaffolds was measured.

### 2.10. Antimicrobial Diffusion Toxicity Test

A disk of the electrospun scaffold (12 mm diameter) was immersed into the bacteria suspension (10 mL) containing 1 mL 10^7^ CFU/mL and 9 mL sterile physiological saline in a 15-mL falcon tube. The falcon tubes were placed on a carousel (IKA, Staufen, Germany) and rotated 360° in the z axis at 30 rpm in an incubator at 37 °C. At the time points 3, 18, 24, 48, 72, and 96 h, 200 μL of phosphate buffered saline (PBS) was removed and serially diluted to 10^−7^ before 100 μL of diluted solutions were added to an agar plate and incubated for 24 h before imaging. The CFUs on plates were counted using the dilution, which was the highest with all bacterial colonies visible as single entities.

### 2.11. Rat Mesenchymal Stem Cell (MSC) Isolation and Passage

MSCs were isolated from four 4–5-week-old male Wistar rat femurs following the method previously described in literature [29,30]. Briefly, the femur was removed from the rats, opened at both ends, and flushed with media. This was then cultured to collect adherent cells, cells from all rats were intermixed, and cells were passaged twice before use. Cells were cultured in minimum essential medium (αMEM) supplemented with 10 U/mL penicillin (Sigma Aldrich, UK), 0.1 mg/mL streptomycin (Sigma Aldrich, UK), 20 mM  alanyl-glutamine (Sigma Aldrich, UK), and 10% v/v fetal bovine serum (FBS; Biosera, Heathfield, UK). Cells were passaged 1:5 using trypsin-EDTA (ethylenediaminetetraacetic acid, Sigma Aldrich) and used below passage 10 for toxicity studies and below passage 5 for differentiation studies.

### 2.12. Noncontact Mammalian Cell Toxicity Study

Toxicity of elution from the electrospun scaffolds was measured by suspending the material over on cultured cells for 24 h. Both primary rat MSCs and the established fibroblast 3T3 cell line were used to test this material. Both cell lines were treated the same in the experiment apart from the rat MSCs were cultured in αMEM as described above and the 3T3 cells were cultured in high-glucose Dulbecco’s Modified Eagle Medium (DMEM) with 10% FBS, both supplemented with 20 mM alanyl-glutamine (Sigma Aldrich, UK) and 0.1 mg/mL of streptomycin (Sigma Aldrich, UK). Cells were trypsinized and added to a 24-well plate (ThermoFisher, Loughborough, UK) at 30,000 cells per well in 1 mL of respective media. The cells were cultured for 24 h in an incubator at 37 °C and 5% CO_2_. Media was replaced and a Transwell^®^ insert (Millicell Hanging Cell Culture Insert, polyethylene terephthalate (PET) 0.4 µm, 24-well) was added to each well containing cells. In triplicate, the following were added to the Transwell^®^ inserts: No scaffold (positive), 1 mL 70% industrial methylated spirit (IMS) (negative), 0% Ag nHA scaffold, 5% Ag nHA scaffold, and 10% Ag nHA scaffold. One mL of media was added to each Transwell^®^ insert and plates were incubated for 48 h at 37 °C and 5% CO_2_. After 48 h the cells within the wells were imaged and metabolic activity was assessed using PrestoBlue™ (ThermoFisher, UK). Media was removed and 700 µL of PrestoBlue™ (10% in complete media) was added to each well and the well plate was incubated for two hours. From the wells, 200 µL of PrestoBlue™ was added in triplicate to a 96-well plate. The fluorescence of the solutions was measured using a plate reader (FLx 800 Bio-Tek Instruments; Thermo Fisher Scientific, UK) with an excitation wavelength of 535 nm and an emission wavelength of 590 nm. The experiment was repeated four times (N = 4, n = 3).

### 2.13. Contact Biocompatibility Study on Rat MSCs

A direct-contact biocompatibility study was used to assess the toxicity of the scaffolds on cells over 21 days. Rat MSCs were used for this study. Disks of each scaffold were added in triplicate to a 24-well plate and 1 mL of media was added to each for 30 min (including a tissue culture plastic [TCP] control). Cells were trypsinized, suspended in media, and then 30,000 cells were added to each well. Well plates were agitated to aid even cell-seeding across the scaffold. Well plates were incubated for 24 h before cell metabolic activity was measured using PrestoBlue™. The scaffolds were transferred to a new well plate before PrestoBlue™ to avoid interference of cells growing on the tissue culture plastic surrounding the scaffold. PrestoBlue™ was measured at 4, 7, 14, and 21 days in culture. N = 2, n = 3.

### 2.14. Live and Dead Cell Staining

Cells were stained with two dyes to identify if the cell was alive or if the cell scaffold was disrupted (dead). Cells were stained using the LIVE/DEAD™ Viability/Cytotoxicity Kit (Thermofisher, UK) as per the instructions. In brief, cells were washed in PBS and then Calcien and Ethidium homodimer-1 (both at 5 µM concentration) were incubated with cells in media for 30 min. Samples were imaged using a fluorescent microscope (Axioplan 2, Zeiss, Jena, Germany) using the built-in filters for FITC (calcein) and TRITC (Ethidium homodimer-1).

### 2.15. Total Sample DNA and Alkaline Phosphatase Production Measurements

Alkaline phosphatase (ALP) is an early indicator of MSC differentiation to osteoblasts and can be measured using an ALP detection test. ALP was normalized to DNA to give a reading of alkaline phosphatase activity per cell regardless of cell number. Media was removed and samples washed three times with PBS. Cell digestion buffer was made up using deionized water and 1% triton X-100 (Sigma, UK). Samples were incubated at 4 °C for 1 h in the cell digestion buffer. The samples were then subjected to two freeze–thaw cycles (–20 °C to room temperature), which included solution agitation by pipetting when thawed to ensure thorough lysis of cells. After the second freeze–thaw cycle, an alkaline phosphatase diethanolamine detection kit was used to assess the alkaline phosphatase activity of the cultures. Then, 40 µL of cell culture lysate solution was placed in a 96-well plate in triplicate with 210 µL alkaline phosphatase reaction buffer. A standard curve of para-nitrophenyl (pNP) product was prepared with concentrations of 250, 100, 50, 10, and 0 nmole.ml^−1^ pNP in the alkaline phosphate reaction buffer. Then, 250 µL of the standard solutions were added in triplicate to the 96-well plate. The para-nitrophenylphosphate (pNPP) substrate solution was prepared by dissolving 6.6 mg pNPP per 1 mL of dH_2_O. Then, 10 µL substrate was then added to each well using a multichannel pipette and the plate was incubated in the dark at 37 °C. The plate was monitored for color change and, at an appropriate time point, the absorbance at 405 nm was read, ensuring that the readings were within the range provided by the standard solutions. The amount of pNP in the sample wells was then calculated using the standard curve and used as a measure of ALP activity.

After a further freeze–thaw cycle, Quant-iT™ PicoGreen^®^ dsDNA assay (ThermoFisher, UK) was used in accordance with the manufacturer’s instructions to quantify the DNA content of the cultures. In detail, 50 µL of cell culture lysate was placed in a 96-well plate in triplicate along with 50 µL dH_2_O. A standard curve of DNA was also prepared in triplicate with final concentrations of 1000, 500, 100, 50, 10, 5, and 0 ng.mL^−1^ when 100 µL of PicoGreen reagent was added to the 100 µL of standard solution in each well. Then, 100 µL of freshly made PicoGreen reagent was added to all the wells using a multichannel pipette. The plate was incubated in the dark for 4 min, after which the fluorescence was read using an excitation wavelength of 485 nm and an emission wavelength of 528 nm. The standard curve was plotted and used to convert the fluorescence units of the wells into the DNA content of the original cultures.

### 2.16. Statistical Methodology

Statistical analysis was conducted using ANOVA run with multiple comparisons using GraphPad Prism 8 (GraphPad Software, Inc., San Diego, CA, USA) with significance indicated with * when *p* < 0.05. No demarcation of lower *p* values was used as all were tested with *p* = 0.05.

## 3. Results

### 3.1. Analysis of Prepared Nanoscale Hydroxyapatite

Nano-hydroxyapatite was prepared via the rapid mixing technique previously published [10]. Using TEM, the nanoscale hydroxyapatite particles were visualized and contained crystalline and amorphous regions, typically less than 100 nm in size. Silver was found within the nanoscale hydroxyapatite particles, with the visible formation of nanoparticles of silver. Both materials showed the peaks associated with hydroxyapatite in XRD patterns. XRD showed the presence of silver phosphate peaks but only for the 10 mol.% silver-containing material (Figure 1a). Despite this, in both materials silver nanoparticles were visible under TEM; EDX confirmed that the darker regions found under the TEM contained silver. See Appendix A for EDX mapping of all elements found in sample Figure 1e. The size distribution of particles’ diameter between nHA and nHA with 10 mol.% silver substituted in was found to be similar, with an average of 37 and 40 nm, respectively (Figure 1f).

### 3.2. Ensuring Encapsulation of nHA by Electrospun Fibers

Both silver and nondoped nHA were electrospun into polycaprolactone (PCL) scaffolds with a thickness of between 150 and 200 µm. SEM images showed incorporation of large particles in the fibers that were not observed in plain PCL scaffolds (Figure 2a). When digested, ICP analysis showed that plain PCL had very low levels of elements found within Ag-nHA (Figure 2b). The electrospun nHA showed the presence of calcium and phosphorus but not silver. Both electrospun scaffolds with silver-containing nHA showed the presence of silver alongside calcium and phosphorus (Figure 2b). This demonstrated that nHA was successfully incorporated into the electrospun fibers.

### 3.3. Investigation of Silver Release and Scaffold Loss in Accelerated Degradation Study

An accelerated degradation assay was performed using sodium hydroxide at 37 °C to understand the degradation profile of the electrospun scaffolds and to observe release profiles (Figure 3). All scaffolds degraded at the same rate with the presence of nHA having no apparent effect on degradation profile (Figure 3a). Cumulative release of silver was observed for the full 30 days (Figure 3b). The plot of release rate (Figure 3c) indicated that in the early period of days 1 to 5 the release was from surface-bound origins, whereas from days 7 onwards the release was from silver within the polymer that was then released through degradation. Mass loss from degradation was nonexistent in the 1- to 7-day period and then rapidly degraded from there, which reinforced the conclusions drawn from the silver release rate.

### 3.4. Antimicrobial Impact of Scaffolds

Both silver-containing scaffolds showed an antibacterial effect on both agar diffusion and on bacterial suspensions. PCL-only scaffolds and those with undoped nHA showed no innate antimicrobial action. Scaffolds containing silver of any concentration showed an antimicrobial response on agar diffusion (Figure 4a,b). Bacterial suspension cultures in PBS exposed to scaffolds for 24 h demonstrated the antimicrobial properties of the scaffolds containing silver. The 10 mol.% silver nHA was studied in a time course assay against bacteria and this showed a significant reduction in bacteria over time (Figure 4e,f). The scaffold reduced the viable bacteria count to undetectable levels by 48 h for *E. coli* and 96 h for *S. aureus*.

### 3.5. Antimicrobial Impact of Scaffolds

Indirect toxicity assays found that higher silver concentrations were more toxic to both cell types while a longer, 21-day direct growth experiment showed a reduced toxic impact of these high silver samples on cells over time. When tested using fibroblast 3T3 cells, all scaffolds had a significantly higher viability than the negative (cytotoxic) control (Figure 5a). However, the scaffold containing 10 mol.% silver was also significantly different to the cell-only positive control, indicating some toxic impact. When the materials were tested on primary rat MSCs the toxic impact was more pronounced. The scaffold with 10 mol.% silver was significantly different to the cell-only positive control, but no statistical difference was found with the negative cell death control (Figure 5b). All the remaining materials were statistically different to the negative cell death controls. A longer, 21-day direct growth experiment found that cell metabolic activity for 5 mol.% silver-containing scaffolds were significantly increased compared to the PCL-only scaffolds (Figure 5c). The 10 mol.% silver-containing scaffolds reduced the metabolic activity when compared to 5 mol.% silver but did not show a toxic impact when compared to the PCL-only scaffold. Live cell morphology can be observed from live fluorescent images (Figure 5d) along with the proportion of live to dead cells. Cells appeared to have a more spread out morphology on the nHA silver-containing material than in the PCL-only control.

### 3.6. MSC Osteogenic Differentiation Study by ALP Quantification

The impact of silver on the differentiation of primary MSCs down the osteoblast linage was measured after 21 days in culture by quantifying ALP content per unit of DNA (Figure 6). ANOVA showed a significant difference with 5 mol.% and the lower silver concentrations in regard to ALP production. Scaffolds with 10 mol.% silver are not significantly different to 5 mol.% silver scaffolds.

## 4. Discussion

Electrospun scaffolds with incorporated silver nano-hydroxyapatite are nontoxic to cells at a concentration that was antimicrobial, making them suitable for use as antimicrobial bone implants. In 2017, Anjaneyulu et al. [25] published a study looking into nanoscale electrospun fibers containing silver-doped hydroxyapatite and its hemocompatibility. Our study took a different direction, producing scaffolds with fibers to allow ingrowth of cells with larger pore sizes having positive impact on osteoblast cell growth [31]. Our study also included an in-depth microbiology study looking at both contact and noncontact tests. Investigating the impact of the scaffolds on bone regeneration via analyzing the toxicity on primary MSCs, for relevant cell toxicity, and 3T3 fibroblasts allowed toxicity comparison between labs.

The rapid manufacturing of nHA pioneered by Wilcock et al. [10] was replicated to produce silver-doped nHA powder. The characterization of nanoscale hydroxyapatite indicated it was comparable to that produced in previous studies. We were able to go one step further by confirming via TEM XRD that the denser (black) areas on the TEM contained silver. This had been hypothesized by the previous paper but not directly confirmed [21]. Under XRD we confirmed that the material was both hydroxyapatite and that it contained silver. TEM analysis showed that the nHA was in a nanoscale particulate form. The lack of an XRD-measurable silver phosphate phase in the 5 mol.% Ag nHA demonstrated that the silver had been incorporated into the nHA, either by substitution of silver into the HA crystal lattice for calcium or by doping of the silver ions onto the HA structure (i.e., not replacing the ions of the HA crystal). This result was expected for our method as our previous reports demonstrated a pure hydroxyapatite phase precipitated at 5 mol.% using this reaction [21].

The silver-doped nHA was successfully electrospun into PCL scaffolds. PCL is a popular material used in bone-related applications as its long-term degradation profile is suited to the slower healing rate of bone compared to soft tissues. Incorporating silver-containing tricalcium phosphate was reported by Schneider et al. [26], who found it effective against *E. coli* but did not test against the more common bone infection-related pathogen *S. aureus*. Jiang et al. [19] assessed antibacterial activity using both of these bacteria with a scaffold formed by in situ foaming, which allowed inclusion of silver/nHA up to 40% of total weight. This technique does not produce scaffolds, however, instead producing a monolithic block that would need to be carved to dimensions required during theater. Anjaneyulu et al. [25] investigated using both bacteria types and manufactured formed scaffolds but did not assess bone cell biocompatibility.

Our scaffolds degraded in an accelerated degradation study and showed a two-phase silver release system. Initially, a rapid burst of silver was released, decreasing in rate until day 7. During the same initial period of degradation, the polymer had not lost any mass. This rapid silver release is likely to arise from surface- (or near surface-) bound material. After the initial 7 days, the material lost mass and silver was released for the remainder of the experiment. Rothstein et al. described this in a model of an initial surface erosion system, which then transitions into bulk erosion [32]. The combination of the two allows for a rapid release when first implanted to clear immediate microorganisms and then a slower sustained release of the remainder of the implant life to prevent colonization from opportunistic bacteria, which may later attach to the device through a hematogenous route. Liu et al. [15] used electrochemical deposition of silver onto their electrospun fibers to provide an initial burst of silver over two days instead of a continual release. This would help prevent potential buildup of silver toxicity but would not prevent later infections from occurring. A continuous release of calcium and phosphorus was detected using ICP, which has been widely reported to stimulate bone regeneration [33]. An implanted material can act as a desirable location for bacteria to colonize with an infection requiring far fewer bacteria to start than in a normal tissue [34]. A continuous silver release would prevent opportunistic bacteria from infecting the implant at a later date. Due to the limited nature of our degradation method we cannot report on the exact amount of silver released per unit of time in vivo, as physiological conditions would impose a more complicated environment that may impact silver release. Despite not knowing the exact amount released, we showed that it is sufficient to prevent bacteria growth in non-accelerated studies.

Antimicrobial tests using both Gram-positive and Gram-negative bacteria showed that an antimicrobial response was found for 2, 5, and 10 mol.% incorporated silver-doped nHA. In a contact agar diffusion assay, both the silver-containing materials were found to produce an area of inhibition, whereas hydroxyapatite by itself does not [35]. Looking at the reduction of bacteria in solution, both bacteria were reduced beyond detection within 96 h. In our study we found silver was far more effective against *S. aureus* than against the *E. coli* bacteria. This was unexpected, as the thicker wall of the Gram-positive *S. aureus* is arguably better defended against the uptake of silver particles than thinner Gram-negative cell walls [36]. Resistance to silver can be acquired by bacteria [37], although it is a relatively broad spectrum medicant. This is significant, as *S. aureus* is among the most common pathogens found in bone infections [1,2]. Recent studies have shown that treating the motile *S. aureus* in an infected 3D tissue model is more challenging than treating a monolayer bacteria assay [38]. Therefore, this would be the logical progression for the testing of this material. One side effect of silver is that it can also be toxic to mammalian cells at elevated concentrations, which we investigated with our scaffolds.

The toxicity of the scaffolds was related to both the silver concentration and on the cell type being tested. Against the standard cell line of mouse fibroblast 3T3 cells there was a small indication of toxicity with scaffolds containing 10 mol.% silver. When using a primary rat MSCs there was clear toxicity detected with this highest concentration of silver. This shows the importance of not just using a single cell line to determine toxicity of a material but also of the correct choice of relevant cells. For a more in-depth review of the different modes of action of silver nanoparticles on various cell types, see Zhang et al. [39]. Using 3T3 cells is a useful method of comparing relative toxicities between different laboratories as they are well known and utilized in international standards for cytotoxicity (ISO 10993-5). The primary MSCs are more difficult to standardize between labs and are, therefore, harder to use to compare toxicity. They do give a more realistic insight on what levels of toxicity might be harmful in the environment the medical device is intended for. From the results presented, 5 mol.% silver-doped material at 20 wt. mol.% to PCL had an antimicrobial impact without being toxic to mammalian MSCs.

Using a longer study of 21 days, the 10 mol.% silver nHA scaffolds maintained a viable and growing cell population on par with the plain PCL material. Hydroxyapatite appeared to boost the metabolic activity of the MSCs by day 21 with the combination of 5 mol.% silver and hydroxyapatite, resulting in even higher levels of cell activity. Increasing the amount of silver to 10 mol.% did not increase the trend of increased proliferation. The initial high release rate from the material was potentially too toxic for the cells early on but, once the release level had fallen to a consistent 50 mol.%, surviving cells would then have been able to proliferate. After this period, the metabolic activity of cell on the 10 mol.% silver nHA scaffold increased at a similar growth rate to other materials. This is highly advantageous in a wound-healing application as it would allow an initially high burst to clear bacteria, which then lowers to allow the mammalian cells to begin the healing process.

Hydroxyapatite is widely reported to increase MSC differentiation into osteoblasts [40,41,42] so we focused on investigating the impact of silver on this process showing increased ALP production. Huge disparity can be found in the literature on whether silver has a significant impact on differentiation of MSCs. Some authors find that silver nanoparticles have no impact on MSC differentiation [43,44]. Other papers report on an impact of silver nanoparticles on MSC differentiation down the osteogenic lineage [45,46]. Conversely, some authors argue that silver particles can cause adipogenic differentiation [47] of MSCs, whereas others have found no impact at all on adipogenic differentiation [48]. One aspect that varies between these papers is the size of the silver nanoparticles, which might be a reason between the widely different reporting of the impact of silver on differentiation. Powers et al. found impact on neurodevelopment from silver on zebrafish PC12 cells, which was very dependent on both silver composition and silver nanoparticle size [49]. Another variation between papers is the wide range of different MSCs that are tested for differentiation, from both different tissue and animal origins. As even MSCs from different tissue origins vary in behavior, it is not surprising that different conclusions have been reported by multiple studies [50]. We found that 5 mol.% silver had the highest ALP production at day 21, although there was no significant difference to 10 mol.% silver. It is possible that either diminishing returns or an already reached optimum concentration of silver is the reason for no further increase at 10 mol.% silver. Another explanation is that the slightly more toxic impact of 10 mol.% silver offsets any additional osteogenic stimulation. Our study demonstrated that silver-doped nHA incorporated into PCL scaffolds may have a positive effect regarding MSC differentiation along an osteoblastic lineage due to increased levels of ALP production.

Limitations to the study include no mapping of elements within the electrospun membrane containing nHA. However, Figure 2b does show evidence that the material is present within the electrospun membranes but the precise location of the nHA is not known. Another limitation is that, while using bacteria typical of infections in bone, neither of the isolates was taken from a patient bone infection. Another limitation on the microbiology side is the simple nature of the tests used; it should be considered that (along with other disparities) osteomyelitis is associated with a hypoxic environment, which was not reflected by our experiments. Further work is needed to address these issues, including the use of a bone infection model to more accurately reproduce the conditions found in vivo. In a recent study we found that a material with antimicrobial impact on planktonic bacteria and simple biofilm models was then not effective on *P. aeruginosa* in tissue-engineered skin models, believed to be due to the bacteria motility [24]. This is something we intend to investigate in future work.

This research found that silver nHA containing scaffolds has the potential to act as an antimicrobial scaffold while stimulating bone tissue regeneration. The scaffolds degrade slowly and release a continuous amount of silver into the local environment over time. They are nontoxic to mammalian cells at a concentration that is toxic to both Gram-negative and Gram-positive bacterial strains. Investigating the impact of silver on MSC differentiation has suggested that higher values of silver promote MSC differentiation.

## 5. Conclusions

Silver-doped nanoscale hydroxyapatite may be incorporated into electrospun PCL scaffolds to fabricate a device that is both nontoxic to cultured mammalian cells and capable of inhibiting the growth of pathogenic microorganisms. The scaffold demonstrated antibacterial activity against both Gram-positive and Gram-negative bacterial strains commonly found in bone infections. The material was shown to release silver in an initial burst followed by a more sustained bulk degradation release. Silver appeared to enhance MSC differentiation down an osteogenic path when used in nontoxic concentrations. Silver-doped nHA containing scaffolds exhibited antimicrobial activity while stimulating bone tissue regeneration, which shows promise as a dual-action medical device to address unmet clinical needs in orthopedic and dental bone surgery.

## Figures and Tables

**Figure 1 jfb-11-00058-f001:**
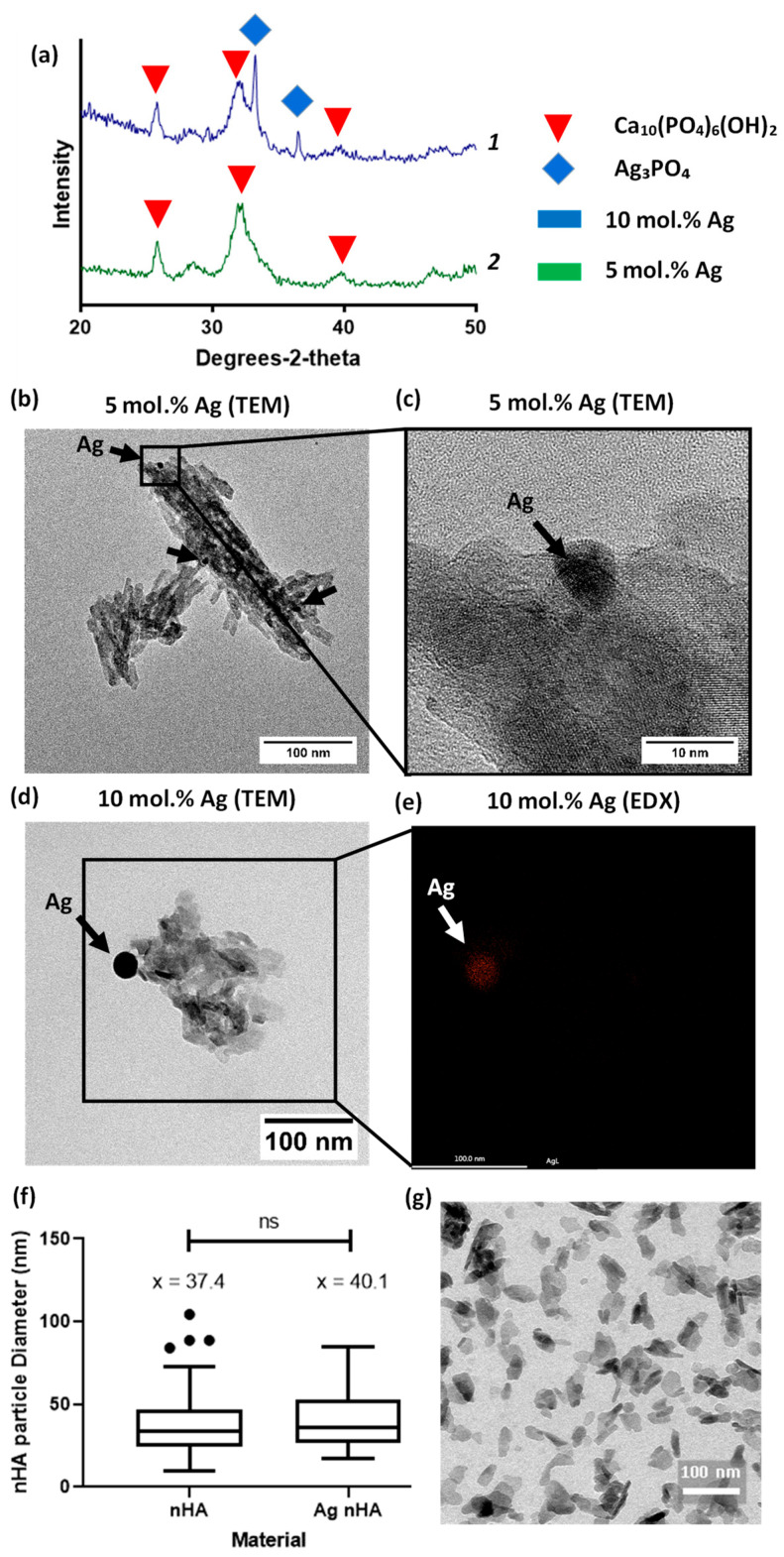
Evaluation of silver-containing nanoscale hydroxyapatite (**a**) XRD patterns of 5 and 10 mol.% silver doped nanoscale hydroxyapatite (Ag nHA), identifying peaks for hydroxyapatite Ca_10_(PO_4_)_6_(OH)_2_ and silver phosphate (Ag_3_PO_4_) phases within the sample. Silver phosphate was only identified in the 10 mol.% silver-containing material. Plot line 1 (blue) is 10 mol.% Ag nHA and plot line 2 (green) is 5 mol.% Ag nHA; (**b**) TEM image of nHA sample containing 5 mol.% silver. Silver deposits visible and highlighted by black arrows; (**c**) TEM at higher magnification on one of the silver deposit areas; (**d**) TEM image of nHA sample containing 10 mol.% silver. Larger silver deposit noted by black arrow; (**e**) Spatial EDX map of location of silver, mapped over the same location as that boxed in (**d**); (**f**) Particle diameter comparison between nHA and nHA with 10 mol.% silver shown as a Tukey box plot, ns = not significant; (**g**) TEM micrograph of nHA nanoscale particles.

**Figure 2 jfb-11-00058-f002:**
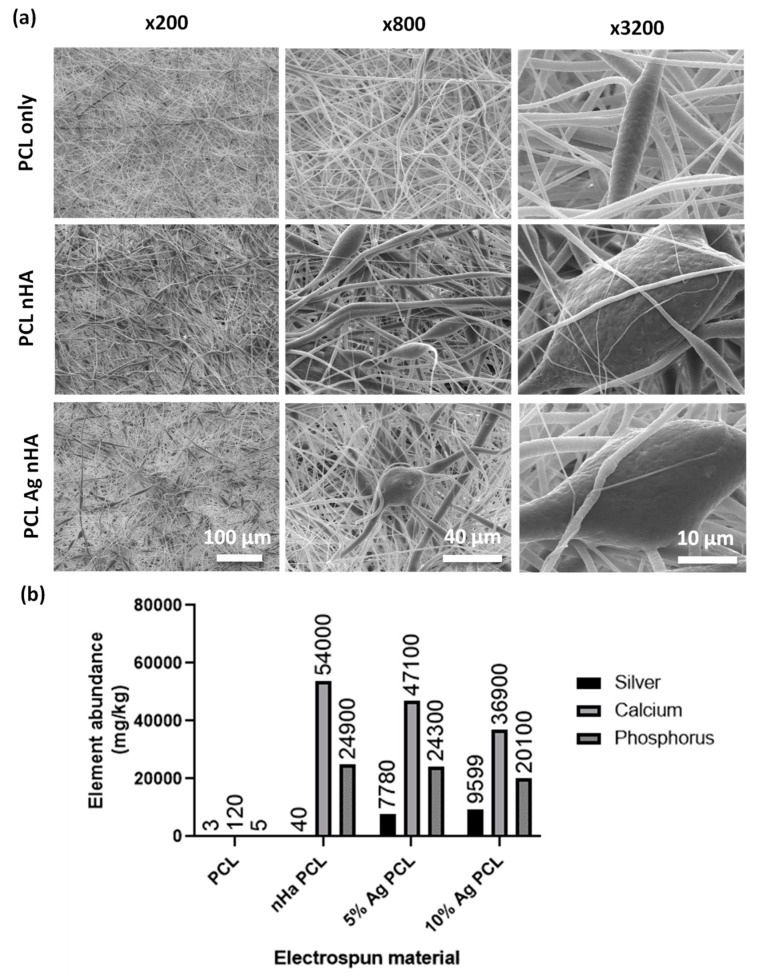
Scaffold fibers containing nHA (**a**) SEM micrographs of electrospun scaffold. Top: Polycaprolactone (PCL) scaffold. Middle: PCL scaffold containing nHA. Bottom: PCL scaffold containing 10 mol.% Ag nHA. Different magnifications were used to visualize encapsulation of nHA particles into the electrospun scaffold; (**b**) Inductively coupled plasma (ICP) analysis of dissolved electrospun scaffolds showing mg/kg of elements present. All % Ag materials are in mol.%.

**Figure 3 jfb-11-00058-f003:**
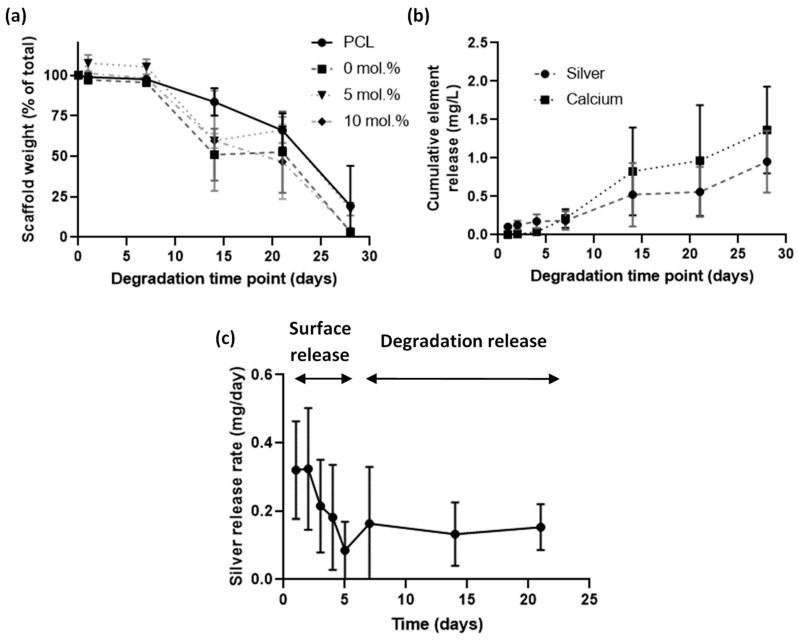
Accelerated degradation study of electrospun scaffolds containing nHA, over a 28-day period in 0.1 M sodium hydroxide. (**a**) Remaining weight over degradation time of different mol.% silver scaffolds; (**b**) Release profile of silver from an electrospun scaffold containing 10 mol.% Ag nHA over a 28-day period; (**c**) Plot of the silver release rate per day. All % Ag materials are in mol.%.

**Figure 4 jfb-11-00058-f004:**
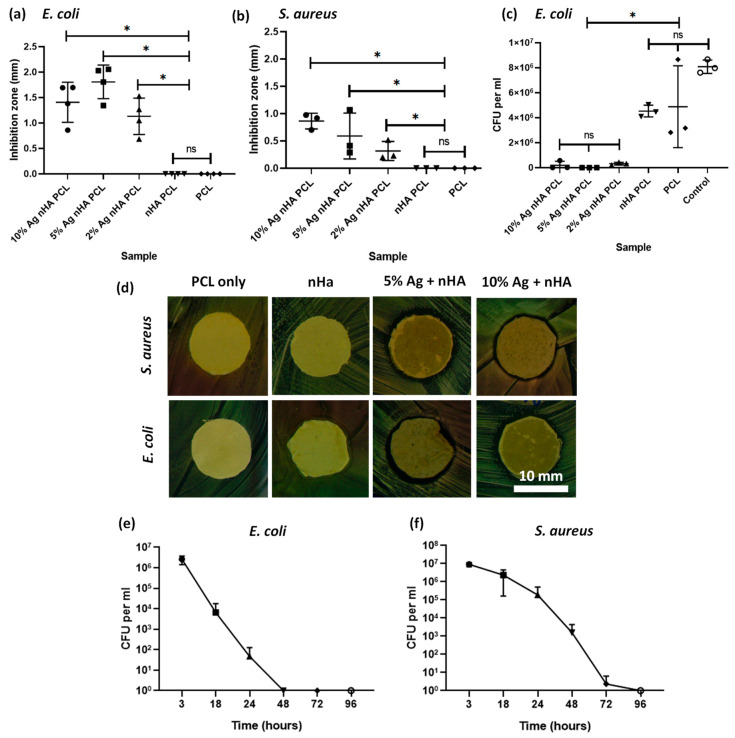
Antibacterial studies on *E. coli* and *S. aureus* bacteria using the electrospun scaffold. (**a**) Agar diffusion test using disks of the electrospun samples against *E. coli* bacteria. Graph displaying individual data plots with a line demarking the mean value; (**b**) Agar diffusion test using disks of the electrospun samples against *S. aureus*; (**c**) Disks of electrospun samples suspended in PBS containing bacteria, measuring viable colonies after 24 h; (**d**) Optical images of the area of inhibition ring surrounding a 12-mm disk of scaffold containing either 10 mol.% silver or plain PCL and tested against *S. aureus* and *E. coli*; (**e**) Bacteria count after *E. coli* are suspended in PBS and exposed to a disk of electrospun 10 mol.% silver nHA for 3, 18, 24, 48, 72, and 96 h; (**f**) Same as (**e**) but using *S. aureus* bacteria. All graphs show mean ± standard deviation. All % Ag materials are in mol.%.

**Figure 5 jfb-11-00058-f005:**
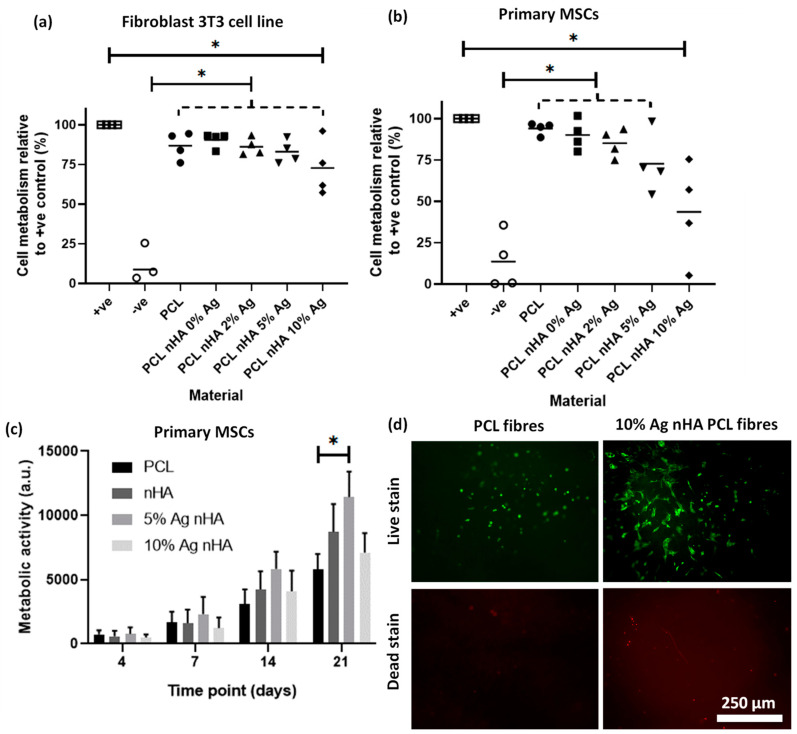
Cell toxicity from the electrospun scaffold with different concentrations of silver nHA assessed by cell metabolic activity via PrestoBlue™ assay. (**a**) A 48-h noncontact toxicity test with the scaffold in the media above a monolayer of 3T3 cells. Graph displaying individual data plots with a line demarking the mean value. Dotted line contains each scaffold under it to be compared to another condition; (**b**) A 48-h noncontact toxicity test with the scaffold in the media above a monolayer of primary rat MSCs. Graph displaying individual data plots with a line demarking the mean value. Dotted line without star contains each scaffold to be compare to another condition; (**c**) Cell metabolism after 21 days of direct culture with the cells seeded on top of the scaffold using metabolic activity at each time point. Statistics (ANOVA) conducted on the day 21 measurements; (**d**) Live/dead staining of MSCs cultured on electrospun scaffolds after 21 days.

**Figure 6 jfb-11-00058-f006:**
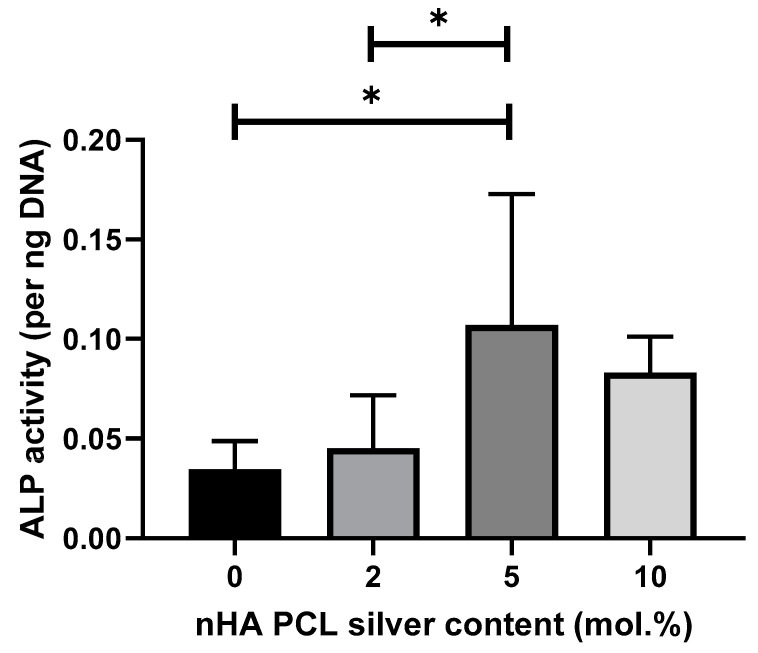
Cell differentiation potential of scaffolds containing different silver contents regarding ALP presence per unit of DNA after 21 days in culture with primary MSCs. Cells were cultured on top of the scaffolds. Graph displaying mean value ± standard deviation. N = 2, n = 3.

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
