# Peer review of "Electrospun Scaffolds Containing Silver-Doped Hydroxyapatite with Antimicrobial Properties for Applications in Orthopedic and Dental Bone Surgery"

_jfb, 2020, doi:10.3390/jfb11030058_

Round 1

Reviewer 1 Report

Electrospun membranes containing silver-doped 2 hydroxyapatite with antimicrobial properties for applications in orthopaedic and dental bone surgery

General comments:

Authors have conceived the work on electrospinning Ag-nHA containing PCL scaffolds and characterizing the morphology, various toxicity, biocompatibility studies including contact and non-contact biocompatibility study.

As this work constitutes various characterization results of prepared nano HA, silver doped nHA along with final electrospun PCL/Ag-nHA fibers, it’s quite ambiguous and deviating from the present proof of concept of underpinning the development of a bioactive electrospun scaffold in biomedical applications and its antimicrobial resistance. For instance, some morphological and structural characterization results of pristine nHA and TEM studies on silver-doped nHA particles had been already reported in previous work of the authors (J. Biomed. Nanotechnol. 2017, Vol. 13, No. 9) and has also been cited.  In order to give the readers a better focus on the characteristics of PCL/Ag-nHA electrospun fibers, authors could consider removing the preliminary characterization results related to pure nHA and Ag-doped nHA mentioned in Figure 1 and Figure 2 to give much clarity.

It is hence suggested to do careful and extensive rearrangement of sections (Section 2 and 3) and revise Figures evidencing the characterization results for providing a better picture to the readers.

How is the nHA separated from the ground powder? All grounded material is nHA?

Sub-sections in Section 2 are not clearly distinguished which makes this section vast and hard to read. Please rectify

In Figure 4 : Figure captions are misleading as Figure 4(a) shows the release profile of nHA containing PCL. Are there no Ag in it? And Figure 4 (b) wherein in degradation tests of PCL containing 10 mol % Ag-nHA has been discussed, did authors check cumulative silver release in all electrospun fibers or please support the basis of including only 10 mol % Ag containing PCL/nHA, whereas in Figure 4 (c) it is not clear of what composition release rate represents.  

Specific comments:

  1. Could authors comment on the mechanical properties of PCL/Ag-nHA fibres and include tensile strength analysis results, as these features are critical for the electrospun nanofibres used for bone and dental regeneration.

  1. It is more relevant if authors avoid using term ‘electrospun membranes’ and use ‘electrospun fibres or scaffolds’ instead, as there are no non-woven being used as support or the prepared electrospun fibres represent non-woven mats/scaffolds either.
  2. What is the rationale behind adding 10 % silver, could authors justify the addition of Ag mol % based on their previous reports
  3. Please include mol % wherever relevant, as it could get confused with wt. % mentioned at various instances in the manuscript.
  4. Figure 4 needs to be revised, for example, Graph in Figure 4(a) is ambiguous, and the data points are not clear to infer.
  5. Page 7, line 280: check for typographical errors and formatting throughout the manuscript, for instance, Ca10(PO4)6(OH)2
  6. What about the thickness of the electrospun scaffolds?

Reviewer 2 Report

Review report

Title: Electrospun membranes containing silver-doped 2 hydroxyapatite with antimicrobial properties for applications in orthopaedic and dental bone surgery

Authors: Thomas E Paterson, Rui Shi, Jingjing Tian, Caroline J Harrison, Mailys De Sousa Mendes, Paul V. Hatton, Zhou Li, Ilida Ortega Asencio

In this manuscript, the authors present the preparation and use of nanoscale hydroxyapatite (nHA) for the development of a bone graft substitute. The manuscript is well written and present interesting results.

In my opinion, this manuscript could be considered for publication, after the following major changes:

  1. The authors should perform a Rietveld analysis of the XRD data in order to better present the structure of the obtained materials.
  2. Furthermore, an EDX spectra of the samples should be added and an EDS ellemnetal mapping of the Ag containing HAp samples should be presented.
  3. In addition to the presented data, the authors should also present XPS studies of the samples.
  4. Regarding the biological assays, some additional discussions and comparison with other reported studies regarding the biological properties of the samples are also required. Please also refer to the following papers:

https://link.springer.com/article/10.1007/s12666-016-1019-0

https://www.hindawi.com/journals/bmri/2013/916218/

https://www.mdpi.com/2079-6412/8/8/276

https://ui.adsabs.harvard.edu/abs/2011LaPhy..21.1265J%2F

https://link.springer.com/article/10.1186/1556-276X-7-324

Reviewer 3 Report

It is an interesting investigation: a rigorous methods was used,  adequate characterization techniques was employed and good analysisis  was presented.

Methodology can be more brief. Statistical analysis was adequate.

See more comments in the paper, to review some details.

Round 2

Reviewer 2 Report

The authors have addressed all the comments but I still think the introduction section and also the conclusions section needs some minor improvement.

Also, the authors should take into consideration to add discussions regarding the existing literature results comparative with their own results.

Author Response

We have improved the introduction by including additional references to back up some statements and to include more of the current literature. We have also re-written sections to allow the text to flow more clearly (62-64, 70 - 75). For the discussion, we have added in some further existing literature results to compare with our own results (394, 417, 421 - 423, 443-443, 452-453). We have restructured key sentences within the conclusions to enhance the readability and removed duplicated statements.